# Antiretroviral Levels in the Cerebrospinal Fluid: The Effect of Inflammation and Genetic Variants

**DOI:** 10.3390/diagnostics13020295

**Published:** 2023-01-12

**Authors:** Jessica Cusato, Valeria Avataneo, Miriam Antonucci, Mattia Trunfio, Letizia Marinaro, Alice Palermiti, Alessandra Manca, Giovanni Di Perri, Jacopo Mula, Stefano Bonora, Antonio D’Avolio, Andrea Calcagno

**Affiliations:** 1Laboratory of Clinical Pharmacology and Pharmacogenetics, Department of Medical Sciences, Amedeo di Savoia Hospital, University of Turin, 10149 Turin, Italy; 2SCDU Infectious Diseases, Amedeo di Savoia Hospital, ASL Città di Torino, 10149 Turin, Italy; 3Unit of Infectious Diseases, Department of Medical Sciences, Amedeo di Savoia Hospital, University of Turin, 10149 Turin, Italy

**Keywords:** pharmacogenetics, ART, CSF, pharmacokinetics, neopterin

## Abstract

Neurocognitive impairments are common in people living with HIV. Some conditions, such as chronic inflammation, astrocyte infection and an impaired blood–brain barrier (BBBi), along with host genetic variants in transporter genes, may affect antiretroviral (ARV) exposure in the cerebrospinal fluid (CSF). The aim of this study was to evaluate ARV CSF penetration according to compartmental inflammation, BBB permeability and single-nucleotide polymorphisms (SNPs) in drug transporter encoding genes. CSF neopterin (ELISA), plasma and CSF ARV concentrations (HPLC) and host genetic variants in *ABCC2*, *HNF4α*, *SLCO1A2* and *SLC22A6* (real-time PCR) were measured. Bi- and multivariate analyses were performed for single ARV and classes. We included 259 participants providing 405 paired plasma and CSF samples. CSF/plasma ratios (CPR) showed an increase for NRTIs and nevirapine with low penetrations for the majority of ARVs. At bi-variate analysis, several associations, including the effect of BBBi (emtricitabine, raltegravir), age (zidovudine and darunavir), and high CSF neopterin (NRTIs and *border-line* for PIs) were suggested. An association was found between genetic variants and integrase strand transfer (*ABCC2* and *HNF4α),* non-nucleoside reverse transcriptase inhibitors (*SLCO1A2*), and protease inhibitors *(SLC22A6).* At multivariate analysis age, gender, BMI, and altered BBB were independent predictors of nucleoside reverse transcriptase CSF concentrations; age (for protease inhibitors) and body mass index and altered BBB (integrase strand transfer inhibitors) were also associated with ARV CSF exposure. We describe factors associated with CSF concentrations, showing that demographic, BBB integrity and, partially, genetic factors may be predictors of drug passage in the central nervous system.

## 1. Introduction

The most recently available UNAIDS data show that approximately 38 million people were living with HIV (PLWH) in 2019 [1]. Notwithstanding, the use of modern highly active antiretroviral treatment (ART) has revolutionized the HIV course and definitely improved patients’ life condition and expectancy, and the onset of HIV-associated neurocognitive diseases (HAND) is still quite common and involves 20–50% of PLWH with significant geographical variability [2,3]. Different grades have been described: the less severe includes asymptomatic neurocognitive impairment (ANI), followed by mild neurocognitive disorders (MND), and finally, by HIV-associated dementia (HAD) [4].

Although it is a very complex and multifactorial disease, several risk factors have been identified, including sustained immune activation that causes (along with HIV viral proteins) structural damage and enhanced permeability of the blood–brain barrier (BBB), poor ART penetration within the central nervous system (CNS) with potential low-level residual viral replication, and a selection of drug resistance, antiretroviral neurotoxicity, patient comorbidities (including vascular disorders), and host genetics [2,4,5]. Other problems, such as altered homeostasis of small hydrophilic compounds or enhanced accumulation of HIV-infected cells within the brain, can arise from the resultant BBB dysfunction [6].

Measuring drug exposure in the central nervous system is highly complex: most of the studies included experimental animals or ex vivo measuring. In vivo techniques using radiological and nuclear assays are ongoing, but so far, cerebrospinal fluid (CSF) concentrations have been used as a proxy of brain tissue exposure. Although recent data in preclinical species suggest that CSF underestimates ART brain tissue concentrations, it may provide helpful insights into the determinants of this process [7].

Few data are available in the literature concerning inflammation, genetic, and BBB-related factors affecting antiretroviral CSF and plasma concentrations. For these reasons, the aim of this study was to measure and characterize ART CSF penetration according to several variables including compartmental inflammation, BBB permeability and single-nucleotide polymorphisms (SNPs) in drug transporter encoding genes.

## 2. Materials and Methods

### 2.1. Study Participants

PLWH receiving antiretroviral therapy enrolled in the PRODIN study (Prot n° 0094178, 30/09/2019, Comitato Etico Interaziendale CIttà della Salute e della Scienza, Turin, Italy) were included. The study was a prospective study that included HIV-positive patients undergoing lumbar puncture for clinical reasons: demographic, clinical, therapeutic variables were collected and serum, genetic and CSF biomarkers were measured. All participants signed a written informed consent form.

Blood samples were collected in lithium-heparin tubes, centrifuged at 3000 rpm for 10 min at 4 °C to obtain plasma, and then stored at −20 °C; CSF was stored at −20 °C.

### 2.2. Biomarkers

Albumin levels were measured in serum and CSF through Immunoturbidimetric methods (AU 5800, Beckman Coulter, Brea, CA, USA). CSAR, intended as a ratio between CSF albumin (mg/L)/serum albumin (g/L), was employed to evaluate BBB function: the definition of BBB damage was derived from age-adjusted Reibergrams (normal if below 6.5 in patients with age <40 years and below 8 in patients with >40 years) [8]. Following CNS inflammation, biomarkers were evaluated: CSF total tau (t-tau, a microtubule-associated protein predominantly expressed in the neurons and associated with taupathologies, such as Alzheimer disease), phosphorylated tau (p-tau, the phosphorilated form which leads to stabilize microtubule assembly), and β- amyloid1-42 (Aβ1-42, produced from amyloid-β precursor protein and accumulated in Alzheimer disease) were quantified by immunoenzymatic methods (Innogenetics) with limits of detection of 87, 15, and 87 pg/mL, respectively. Neopterin, a marker of cellular immune system activation and CNS inflammation, was determined through validated ELISA methods (DRG Diagnostics). Reference values were as follows: t-tau [<300 pg/mL (in patients aged 21–50), <450 pg/mL (in patients aged 51–70), or <500 pg/mL in older patients], p-tau (<61 pg/mL), 1–42 beta amyloid (>500 pg/mL), and neopterin (<1.5 ng/mL).

Circulating HIV RNA was quantified by a real-time polymerase chain reaction (PCR) assay CAP/CTM HIV-1 vs. 2.0 (CAP/CTM, Roche Molecular System, Branchburg, NJ, USA; detection limit: 20 copies/mL of HIV-1 RNA). CSF escape was defined as CSF HIV RNA above 50 copies/mL in patients with plasma HIV RNA below 50 copies/mL or as CSF HIV RNA 1 log_10_ higher than plasma HIV RNA in patients with a detectable plasma viral load. HAND was diagnosed according to the Frascati criteria.

### 2.3. Pharmacokinetic Analyses

Drug concentrations were evaluated through Therapeutic Drug Monitoring (TDM), which is the clinical practice performed on plasma and CSF samples with different extraction procedures in order to avoid toxic effects or therapeutic failures. Drugs were divided in the following classes: INSTIs (Integrase Strand Transfer Inhibitors: raltegravir [RAL, therapeutic range 40 ng/mL], dolutegravir [DTG, therapeutic range 300–2138 ng/mL], and elvitegravir [EVG, therapeutic range 45–2400 ng/mL]), PIs (Protease Inhibitors: atazanavir [ATV, therapeutic range 150–850 ng/mL], darunavir [DRV, therapeutic range 550–7242 ng/mL], amprenavir [APV, therapeutic range 400 ng/mL], lopinavir [LPV, therapeutic range 1000–8000 ng/mL] and ritonavir [RTV, no range ng/mL]), NNRTIs (Non-Nucleoside Reverse Transcriptase Inhibitors: efavirenz [EFV, therapeutic range 1000–4000 ng/mL], etravirine [ETV, therapeutic range 160 ng/mL in naϊve patients, 3000 ng/mL in experienced; 3000 ng/mL], rilpivirine [RPV, therapeutic range 12 ng/mL in naϊve patients, 50 ng/mL in experienced; 500 ng/mL] and nevirapine [NVP, therapeutic range 3000–6000 ng/mL]) NRTIs (Nucleoside Reverse Transcriptase Inhibitors: zidovudine [AZT, no range available], lamivudine [3TC, no range available], abacavir [ABC, no range available], emtricitabine [FTC, no range available] and tenofovir [TDF, therapeutic range 40–180 ng/mL]), plus maraviroc [MVC, therapeutic range 50 ng/mL] and cobicistat [COBI, no range available] [9]. All plasma samples were analyzed basing on a previously published method, and fully validated following FDA guidelines [10]. Concerning CSF, the extraction protocol was modified specifically for CSF (as previously performed for other antiretroviral and antibacterial drugs, and fully validated following FDA and EMA guidelines (as for the plasma method), as a rule of our certified laboratory (UNI EN ISO 9001 and 13485) [10,11,12,13,14,15]. All the analyses were carried out using high-performance liquid chromatography coupled with a tandem mass spectrometry (UHPLC-MS/MS) system.

### 2.4. Pharmacogenetic Analyses

Genomic DNA was isolated from blood samples (MagNA Pure Compact, Roche, Monza, Italy). Genotypes were assessed through a real-time PCR allelic discrimination system (LightCycler 96, Roche, Monza, Italy). Investigated gene SNPs were *ABCB1* (encoding the P-glycoprotein) 3435 C > T (rs1045642), *ABCB1* 1236 C > T (rs1128503), *ABCB1* 2677 G > T (rs2032582), *ABCC2* (encoding the multidrug resistance protein 2)-24 G > A (rs717620), *ABCG2* (encoding the ATP binding cassette subfamily G member 2) 421 C > A (rs2231142), *ABCG2* 1194 + 928 C > A (rs13120400), *HNF4a* (encoding the hepatocyte nuclear factor 4 *a*) 975 C > G (rs1884613), *SLCO1A2* (encoding the solute carrier organic anion transporter family member 1A2) 38 A > G (rs10841795), *SLCO1A2* 516 A > C (rs11568563) and *SLC22A6* (encoding the solute carrier family 22 member 6) 453 G > A (rs4149170).

### 2.5. Statistical Analyses

Patients’ demographic and clinical data have been collected and described as follows: categorical variables were described as frequency and percentage, while numerical ones as the median and interquartile (IQR) range.

Statistical analyses have been focused on investigating the strength of the association between CSF and plasma drug concentrations by calculating the Spearman coefficient and considering the level of statistical significance (*p* value < 0.05). Non-parametric tests of Kruskal–Wallis (for more than two groups) or Mann–Whitney (for two groups) have been used in order to identify differences in continuous variables among different groups of patients, due to the non-normal distribution of data.

Pharmacogenetic analyses were performed considering drug classes, since a single drug sample size was small.

Stepwise multivariate logistic regression analysis was performed in order to analyze which factors are able to predict altered neopterin.

All analyses were performed through the IBM SPSS software, version 26.0 (Chicago, IL, USA).

## 3. Results

### 3.1. Patient Description

We included 259 participants. The majority of patients were male (*n* = 185; 71.4%) with European ancestry (*n* = 194; 74.9%) based on self-reporting (no region level was available); the median age was 48 years (IQR—inter quartile range 42–55) and median BMI was 22.6 kg/m^2^ (IQR 20–25.1). The number of prescribed drugs ranged from 1 to 5, with the large majority of patients (*n* = 180) taking three antiretrovirals. The median CD4 and nadir CD4 cell count were, respectively, 336 (IQR 147–590) and 99 (IQR 25–211) cells/µL. Among the study participants, 47.5% and 56.4% showed plasma and CSF viral loads lower than 50 copies/mL. Markers indicating an altered BBB were found in 19.7% of participants (*n* = 51).

### 3.2. Pharmacokinetic Evaluation

Study subjects provided 405 paired plasma and CSF samples. ABC, RPV, ATV, EVG showed a very good correlation (ρ > 0.70) between plasma and CSF concentrations (Figure 1), but the latter one was lower than the plasma one; several CSF samples had undetectable antiretroviral concentrations (Table 1).

The CSF/plasma ratio (CPR) was calculated for each drug and expressed as a percentage: ABC and AZT showed the highest CSF-to-plasma ratios (CPR) (92% and 97% CPR, respectively), followed by 3TC, FTC and NVP (37%, 39% and 45% CPR, respectively). All the other drugs showed very low CPRs (Figure 2).

We then assessed the effect of several variables on CSF concentrations and CPRs using bivariate and multivariate analysis. In participants with impaired BBB, the following drugs showed significantly higher CSF concentrations: FTC (90.5 vs. 132, *p* = 0.006) and RAL (30.8 vs. 58 ng/mL, *p* = 0.023; we observed higher FTC CPRs in participants with impaired BBB (55.3 vs. 37.2%, *p* = 0.004). Inflammation seemed to have an impact on drug concentrations: NRTIs (*p* = 0.008); in particular, TDF (*p* = 0.038) and FTC (*p* < 0.001) showed higher CSF concentrations in patients with altered neopterin levels (>1.5 ng/mL, Figure 3). A linear correlation was shown for CSF FTC and neopterin levels (*p* = 0.002, ρ = 0.333). Age was directly associated with CSF NRTI exposure (rho = 0.14, *p* = 0.016) with AZT (rho = 0.65, *p* = 0.021) and 3TC (rho = 0.13, not significant *p*-value) being higher in the CSF of elderly participants; however, DRV (rho = 0.27, *p* = 0.018), ATV and RAL (rho = 0.14 and 0.17, respectively, not significant *p*-values) also showed a similar trend. Only for PIs, the CPR results were slightly associated with age (rho = 0.20, *p* = 0.023).

### 3.3. Pharmacogenetic Evaluation

Pharmacogenetic analyses (Figure 4) showed that INSTI concentrations were associated with *ABCC2* and hepatocyte nuclear factor 4 alpha (*HNF4α*) genes that encode for MRP2 and HNF4α, two proteins involved in drug transport and regulation, respectively.

*ABCC2* -24 AA carriers showed significantly lower (*p* = 0.017) INSTI CSF concentrations, whereas *HNF4α* 975 CC genotype carriers had significantly higher (*p* = 0.005) INSTI’s CPR. NNRTI concentrations were affected by *SLCO1A2* polymorphisms: *SLCO1A2* 38 AA carriers showed significantly higher (*p* = 0.021) NNRTI CSF concentrations. Finally, PI’s CPR could have been influenced by the *SLC22A6* 453 variant: genotype GG carriers had significantly lower (*p* = 0.047) PI CPRs. Considering single drugs, genetic polymorphisms’ influence was analyzed (Table 2).

Separate linear regression analyses were performed, including the several variables and the studied genetic variants. Sensitivity analysis assessed the best models for each drug class. Age, gender, BMI, BBBi and ABCB1 for NRTI CSF PK, age for PI CSF PK and CPR, BMI and altered BBB for INSTI CSF PK remained in the final models. No multivariate model was significantly associated with NNRTI CSF concentrations or CPRs. Figure 4 shows the cerebrospinal fluid (CSF, A for integrase inhibitors, INSTI and B for non-nucleoside reverse transcriptase inhibitors, NNRTI) and cerebrospinal fluid/plasma ratio (C for INSTI and D for protease inhibitors, PI) concentrations of different antiretroviral drugs according to single-nucleotide polymorphism.

## 4. Discussion

In the ART era, it is crucial to find and maintain the best treatment option for each patient in order to achieve and maintain optimal viral suppression and to improve patients’ quality of life. In the context of personalized medicine, TDM and pharmacogenetics are precious tools that, together, can identify suboptimal therapies [16].

Several factors affecting drug concentrations in plasma and CNS penetration were investigated in this study. Even though recent data have suggested that CSF is not a good surrogate for brain concentrations in preclinical species, it is the only in vivo biomarker available, and available data suggest a proportionality of CSF penetration of brain tissue exposure [7]. Lumbar punctures are, however, invasive procedures, and alternative surrogate matrices coupled with modeling methods are warranted [17,18]: in this study, CSF was compared to plasma, which is the most widely used standard matrix, in order to evaluate potential correlations. We found that some drugs (ABC, RPV, ATV, EVG) showed a good correlation between plasma and CSF concentrations: considering the high frequency of undetectable CSF drug concentrations due to active transport and BBB function, it is necessary to estimate variables associated with CNS penetration.

We described ARV distribution in plasma and CSF in HIV-affected patients according to genetics and other patient-related features. In these analyses, NRTIs seem to easily cross the BBB reaching the CSF, especially for ABC and AZT (CPRs higher than 90%). This confirms what was found by Ene et al., who suggested that NRTIs as a class have the advantage of good CSF concentration [19]. A study by Strazielle and colleagues [20] demonstrated the increasing potential penetration in CSF of 3TC, didanosine, stavudine and AZT; this was consistent with our results underlying ABC and AZT’s strongest tendency to cross the BBB, followed by 3TC, FTC and NVP, maybe due to their similar molecular characteristics. Other studies correlated CSF drug penetration with the physical and chemical properties of each molecule, underlying the importance of parameters such as molecular weight, protein binding, and lipid solubility. We decided to focus on the role of factors associated to BBB integrity, inflammation and, to a lesser extent, age. In our study, impaired BBB was associated with higher CSF FTC and RAL concentration, whereas CPR increased for FTC (due to increased CSF levels).

Concerning inflammation, NRTI CSF exposure was major in individuals with neuroinflammation, intended as neopterin levels higher than 1.5 ng/mL. Probably, this could be due to tight junction disruption caused by neuroinflammation [21,22], and consequently, small molecules, such as NRTIs, could pass through the BBB in an easier way. Inflammation is able to modulate drug-metabolizing enzymes and transporters contributing to intra- and inter-individual variability of drug exposure.

In particular, inflammatory mediators, through transcriptional and post-transcriptional mechanisms, lead to enzyme and transporter inhibition. Consequently, inflammation has an effect on pharmacokinetic parameters of most of the drugs which are metabolized and transported by these proteins and for which TDM is recommended, in order to improve the personalization of drug treatment for each patient [23].

Age was also associated with an increase in CSF NRTI concentrations, and DRV, RAL and ATV also showed a similar trend, but without statistical significance; this could be attributable to physiological ageing, leading to a more permeable BBB. Advancing age is characterized by different organ impairment: in detail, a reduction in renal and hepatic function could lead to pharmacokinetic changes, including an increase in the volume of distribution of lipid soluble drugs, thus prolongating the elimination half-life; on the other hand, pharmacodynamic changes could involve altered (often increased) sensitivity to several classes of drugs [24].

Reduced homeostatic ability affects different regulatory systems in different subjects, thus explaining, at least partly, the increased interindividual variability occurring as people get older. Important pharmacokinetic and pharmacodynamic changes occur with advancing age. This is one of the few studies showing an effect of host genetics on CSF penetrations of ARV [25].

*ABCC2* 24 C > T affects ABCC2 transcriptional regulation [26]: particularly, the variant seems to have increased activity regarding drug efflux [27]. Studies have shown that the T allele is related to an increased risk of anti-HIV drugs (ATV, LPV, RTV and TDF) toxicity [28]. In addition, the CT + TT group is not able to affect TDF levels as compared to genotype CC [29]. In this study, TT carriers decreased INSTI concentrations compared to other genetic groups—this was probably due to the genetic variant leading to an increased drug efflux from CSF towards the plasma compartment. The *HNF4α* 975 C > G genetic variant is present upstream of the HNF4α promoter gene, and it has no known function [30,31]. Recently, our group suggested the *HNF4α* 975 CC genotype correlated with increased EFV exposure [30]. Moreover, the CG genotype was associated with lower RAL CSF and CPRs concentrations; in these analyses, we confirm these data finding higher INSTI CPRs in CC genotype patients compared to CG/GG ones [13]. *SLC1A2* 38 T > C genetic polymorphism in exon 1 causes an isoleucine to threonine amino acid change, increasing the uptake of estrone sulfate and methotrexate [31]. Furthermore, another study demonstrated that this SNP does not affect RAL levels [13]. In our work, we demonstrated lower NNRTI CSF exposure in TC/CC genotype subjects. Concerning *SLC22A6* 453 G > A, RAL concentrations resulted in not being influenced by this genetic variant [13]; moreover, the T allele was not related to nephrotoxicity in TDF-treated patients. Here, a slightly lower PI CPR was found in GG patients.

This study investigated the role of different factors influencing CSF and plasma ARV concentrations.

In conclusion, considering the complexity of drug-related side effects or viral resistances, it could be beneficial to use an integrated approach, combining both clinical screening, TDM and pharmacogenetic analyses; this combined method could be useful to improve tailored medicine in order to better manage these kinds of patients.

## Figures and Tables

**Figure 1 diagnostics-13-00295-f001:**
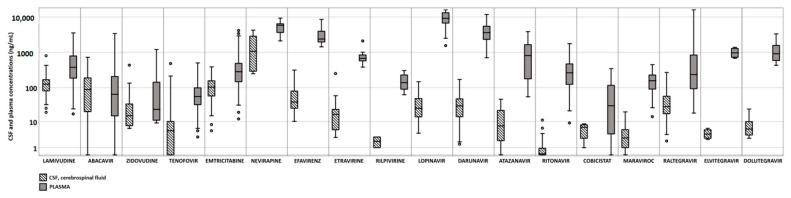
Cerebrospinal fluid and plasma concentrations of different antiretroviral drugs.

**Figure 2 diagnostics-13-00295-f002:**
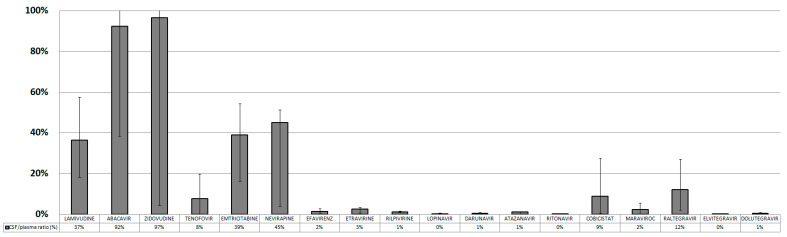
Cerebrospinal fluid/plasma ratio of different antiretroviral drugs.

**Figure 3 diagnostics-13-00295-f003:**
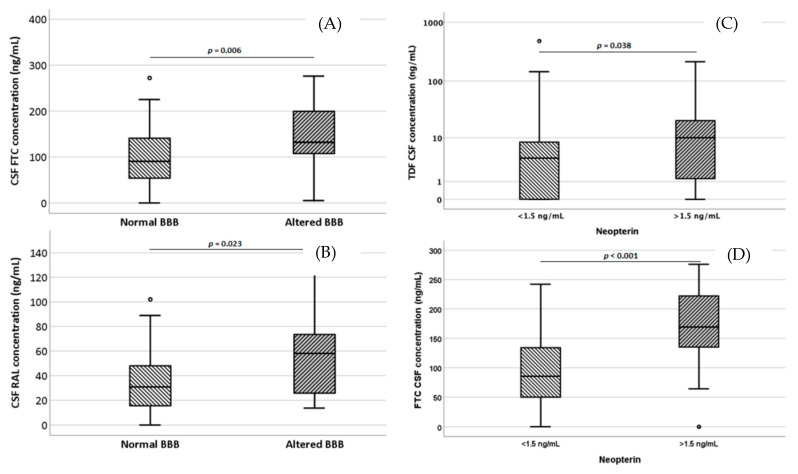
Cerebrospinal fluid according to an altered blood–brain barrier (**A**,**B**) and increased neopterin levels (**C**,**D**). FTC: emtricitabine, RAL: raltegravir, TDF: tenofovir. Circles indicate “out” values.

**Figure 4 diagnostics-13-00295-f004:**
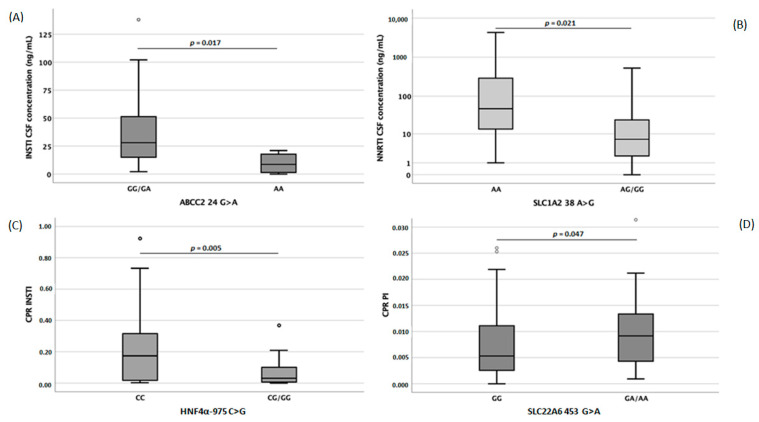
Cerebrospinal fluid (CSF, (**A**) for integrase inhibitors, INSTI and (**B**) for non-nucleoside reverse transcriptase inhibitors, NNRTI) and cerebrospinal fluid/plasma ratio ((**C**) for INSTI and (**D**) for protease inhibitors, PI) concentrations of different antiretroviral drugs according to single-nucleotide polymorphism.

**Table 1 diagnostics-13-00295-t001:** Median CSF drugs’ concentrations and percentages of patients with concentrations above the instrumental limit of quantification (LOQ) and/or limit of detection (LOD); n.c. = not calculable.

	3TC	ABV	AZT	TDF	FTC	NVP	EFV	ETV	RPV	LPV	DRV	ATV	RTV	COBI	MVC	RAL	ELV	DTG
Number of samples	57	34	12	107	104	9	14	14	6	28	78	31	116	4	21	80	4	12
Median CSF conc. (ng/mL)	122.0	87.5	15.0	5.0	102.5	1062.0	40.5	16.5	2.0	25.0	29.4	7.2	0.36	6.5	2.2	30.4	3.9	6.5
IQR	74.5–162	17.5–234	6.7–34.7	0–10	57.2–155	268.5–3050.5	24.6–81.7	5.4–23.4	1–3	13–49	14–47.2	1.9–21.7	0–0.9	2.2–8.1	0.4–5.4	18.7–56.5	2.5–5.7	3.3–11
Percentage of patients with CSF conc. < LOQ	0	14.7	33.3	31.8	8.7	11.1	14.3	28.6	0	60.7	37.2	61.3	94.8	25	71.4	5	75	83.3
Percentage of patients with CSF conc. < LOD	0	11.8	16.7	31.8	6.7	11.1	0	0	0	7.1	2.6	12.9	28.4	0	14.3	1.3	0	8.3
Correlation with plasma (Spearman’s ρ)	ρ = 0.47; *p* < 0.001	ρ = 0.86; *p* < 0.001	ρ = 0.34; *p* = 0.278	ρ = 0.54; *p* < 0.001	ρ = 0.59; *p* < 0.001	ρ = 0.32; *p* = 0.406	ρ = 0.63; *p* = 0.021	ρ = 0.21; *p* = 0.473	ρ = 0.84; *p* = 0.038	ρ = 0.04; *p* = 0.857	ρ = 0.49; *p* < 0.001	ρ = 0.79; *p* < 0.001	ρ = 0.36; *p* = n.c.	ρ = 0.80; *p* = 0.2	ρ = 0.12; *p* = 0.623	ρ = 0.47; *p* < 0.001	ρ = 1; *p* = n.c.	ρ = −0.1; *p* = 0.749

**Table 2 diagnostics-13-00295-t002:** Influence of single-nucleotide polymorphisms on plasma (PL) and cerebrospinal fluid (CSF) drug concentrations. BBB and altered neopterin were associated (*p* = 0.003, Pearson coefficient 9.124).

		3TC	ABV	AZT	TDF	FTC	NVP	EFV	ETV	RPV	LPV	DRV	ATV	RTV	COBI	MVC	RAL	ELV	DTG
* **ABCB1** *	rs1045642																		
	rs1128503				PL 0.022						CSF 0.028 PL 0.035								
	rs2032582			PL 0.041		CSF 0.036	CSF 0.044						CSF 0.031 PL 0.029						
* **ABCC2** *	rs717620												CSF 0.041 PL 0.047				CSF 0.030		
* **SLC22A6** *	rs4149170																		
* **SLCO1A2** *	rs10841795																		
	rs11568563																CSF 0.032		
* **ABCG2** *	rs2231142				PL 0.009														
	rs13120400																CSF 0.017		
* **HNF4α** *	rs1884613																		
* **ABCC10** *	rs9349256																		

## Data Availability

Data are available on request by the corresponding author.

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
