# Peer review of "Antiretroviral Levels in the Cerebrospinal Fluid: The Effect of Inflammation and Genetic Variants"

_diagnostics, 2023, doi:10.3390/diagnostics13020295_

Round 1
Reviewer 1 Report
The authors present a prospective study that included 259 HIV-positive patients undergoing lumbar puncture for clinical reasons. They analyzed the role of different factors influencing cerebrospinal fluid and plasma antiretrovirals concentrations.
The information is complete and concise. The conclusions are consistent with the evidence and arguments presented and address the main question posed. And the references are appropriate.
There are few data reported on inflammation, genetics, and blood-brain barrier-related factors that affect cerebrospinal fluid and plasma antiretroviral drug concentrations. Therefore, this study could be useful to improve medical actions in these patients.
Author Response
R: thank you for your revision! Results have been improved, as you recommended.
Reviewer 2 Report
Manuscript ID: diagnostics-2089403
Title: Antiretroviral Levels in the Cerebrospinal Fluid: The Effect of Inflammation and Genetic Variants
Authors: CUSATO et al.
In this manuscript, Cusato et al. investigate the role of different factors influencing CSF and plasma antiretroviral concentrations. The Authors evaluated markers of inflammation, permeability of the BBB and SNPs related to drug transporter genes. They found associations on how demographic characteristics like age, changes in the BBB and some genetic variants may affect the way drugs can pass into the CNS. They conclude that a combined approach using clinical screening, therapeutic drug monitoring and genetic studies would be of benefit to provide better clinical management tailored to each patient.
Overall, this is an interesting manuscript and provides valuable scientific information to advance the field. The Introduction contains enough information to provide context and relevance of the study. The methods used to characterize the biomarkers, drug concentrations and the genetic analysis seem well considered. However, in the part of Materials and Methods some clarifications and corrections are needed. The results are presented thoroughly in the manuscript and supplementary files. The interpretation of the results is adequate and consistent with the research findings.
However, the Reviewer found some issues in the manuscript and would like to see them addressed by the Authors before this manuscript is ready to be considered for publication.
Specific comments and suggestions:
INTRODUCTION
Line 22: “astrocyte infecton”. Please correct typo.
Line 40: “HAART”. This may be changed to “ART” that currently is most used. Please verify this in all the manuscript.
Line 44.“ HIV (PLWH) in 2019 (Webpage).” Please add the citation on the text and in the Reference section (number 33) please add the complete citation, the web address and when it was accessed.
METHODS
Line 106. “2.3. Pharmacokinetic analyses” section. Please add in the Methods section which are the expected therapeutic concentrations (for the TDM) for each of the drugs being measured. The ratios between plasma and CSF indicate which compartment have a higher or lower concentration but without the expected values it is not clear whether they are within therapeutic range or not.
Line 125. “2.4. Pharmacogenetic analyses” section. Please add in the Methods section which proteins are encoded or associated with the gene SNPs being studied.
RESULTS
Line 152. “European ancestry”. Please explain whether the patients are classified as being of “European ancestry” based on self-reporting or was it determined by genetic testing. Also, can this classification be narrowed down to region level, such as which region in Italy, or by ethnic origin since allele frequencies may vary among different groups.
Line 168. Fig 1. And Line 174. Fig 2. The size of the letters is too small to be read easily. Please consider using a larger front or increasing the size of the figures.
OTHERS
Line 300. Please remove section 6. Patents if it does not apply to your work.
Line 305-306. “Author Contributions”, Please remove the sentence “For research articles with several authors, a short paragraph specifying their individual contributions must be provided.”
Lines 312 to 314. Please remove the sentence “Please turn to the CRediT taxonomy for the term explanation. Authorship must be limited to those who have contributed substantially to the work reported.”
Thank you kindly.
Author Response
R: thank you for your revision.
INTRODUCTION
Line 22: “astrocyte infecton”. Please correct typo.
R: thanks. The typo has been corrected.
Line 40: “HAART”. This may be changed to “ART” that currently is most used. Please verify this in all the manuscript.
R: thanks. The acronym has been changed, as you recommended.
Line 44.“ HIV (PLWH) in 2019 (Webpage).” Please add the citation on the text and in the Reference section (number 33) please add the complete citation, the web address and when it was accessed.
R: thank you for your comment. The typo has been corrected.
METHODS
Line 106. “2.3. Pharmacokinetic analyses” section. Please add in the Methods section which are the expected therapeutic concentrations (for the TDM) for each of the drugs being measured. The ratios between plasma and CSF indicate which compartment have a higher or lower concentration but without the expected values it is not clear whether they are within therapeutic range or not.
R: thanks for the comment. Therapeutic ranges have been added, if available, also adding the reference citation.
Line 125. “2.4. Pharmacogenetic analyses” section. Please add in the Methods section which proteins are encoded or associated with the gene SNPs being studied.
R: proteins have been added, as you suggested.
RESULTS
Line 152. “European ancestry”. Please explain whether the patients are classified as being of “European ancestry” based on self-reporting or was it determined by genetic testing. Also, can this classification be narrowed down to region level, such as which region in Italy, or by ethnic origin since allele frequencies may vary among different groups.
R: thank you for the comment. The information is obtained by self-reporting and this has been added to the manuscript. Unfortunately, the region level is not available.
Line 168. Fig 1. And Line 174. Fig 2. The size of the letters is too small to be read easily. Please consider using a larger front or increasing the size of the figures.
R: thanks. The size has been increased.
OTHERS
Line 300. Please remove section 6. Patents if it does not apply to your work.
R: thanks for your revision. The section has been removed, according to your suggestion.
Line 305-306. “Author Contributions”, Please remove the sentence “For research articles with several authors, a short paragraph specifying their individual contributions must be provided.”
R: thanks for your comment. The sentence has been removed, as recommended.
Lines 312 to 314. Please remove the sentence “Please turn to the CRediT taxonomy for the term explanation. Authorship must be limited to those who have contributed substantially to the work reported.”
R: thanks for your suggestion. The sentence has been removed, according to your comment.